# Persistence of chikungunya ECSA genotype and local outbreak in an upper medium class neighborhood in Northeast Brazil

**Jaqueline Goes de Jesus[1©], Gabriel da Luz Wallau[2©], Maricelia Lima Maia[3,4],
Joilson Xavier[5], Maria Aparecida Oliveira Lima[3], Vagner Fonseca[5,6], Alvaro Salgado de
Abreu[5], Stephane Fraga de Oliveira Tosta[5], Helineide Ramos do Amaral[3], Italo Andrade
Barbosa Lima[1], Paloma Viana Silva[1], Daiana Carlos dos Santos[1], Aline Sousa de
Oliveira[1], Siane Campos de Souza[1], Melissa Barreto Falcão[4], Erenilde Cerqueira[3],
Laís Ceschini Machado[2], Mariana Carolina Sobral[2], Tatiana Maria Teodoro Rezende[2],
Mylena Ribeiro Pereira[7], Felicidade Mota Pereira[8], Zuinara Pereira Gusmão Maia[8],
Rafael Freitas de Oliveira França[7], André Luiz de Abreu[9], Carlos Frederico Campelo de
Albuquerque e Melo[10], Nuno Rodrigues Faria[11], Rivaldo Venâncio da Cunha[12,13],
Marta Giovanetti[5,14©] \*, Luiz Carlos Junior Alcantara[5,14] \***

1 Laboratório de Patologia Experimental, Instituto Gonçalo Moniz, Fundação Oswaldo Cruz, Salvador, Brazil,
2 Departamento de Entomologia, Instituto Aggeu Magalhães, Fundação Oswaldo Cruz, Recife, Brazil,
3 Universidade Estadual de Feira de Santana, Feira de Santana, Brazil, 4 Secretaria de Saúde de Feira de
Santana, Ministério da Saúde, Feira de Santana, Brazil, 5 Laboratório de Genética Celular e Molecular, ICB,
Universidade Federal de Minas Gerais, Belo Horizonte, Minas Gerais, Brazil, 6 KwaZulu-Natal Research
Innovation and Sequencing Platform (KRISP), College of Health Sciences, University of KwaZulu-Natal,
Durban, South Africa, 7 Departamento de Virologia, Instituto Aggeu Magalhaes, Fundação Oswaldo Cruz,
Recife, Brazil, 8 Laboratório Central de Saúde Pública da Bahia, Salvador, Bahia, Brazil, 9 Secretaria de
Vigilância em Saúde, Coordenação Geral de Laboratórios de Saúde Pública, Ministério da Saúde, Brasília,
Brazil, 10 Organização Pan-Americana da Saúde/Organização Mundial da Saúde—(OPAS/OMS), Brasília,
Brazil, 11 Department of Zoology, University of Oxford, Oxford, United Kingdom, 12 Departamento de
Clínica Médica, Faculdade de Medicina, Universidade Federal do Mato Grosso do Sul, Campo Grande,
Brazil, 13 Fundação Oswaldo Cruz, Rio de Janeiro, Brazil, 14 Laboratório de Flavivírus, Instituto Oswaldo
Cruz, Fundação Oswaldo Cruz, Rio de Janeiro, Brazil

© These authors contributed equally to this work.
\* luiz.alcantara@ioc.fiocruz.br (LA); giovanetti.marta@gmail.com (GM)

pone.0226098

biomedico, ITALY

**Data Availability Statement:** Data are available in
S1 Table and sequences deposited in GenBank

## Abstract

The chikungunya East/Central/South/Africa virus lineage (CHIKV-ECSA) was first detected
in Brazil in the municipality of Feira de Santana (FS) by mid 2014. Following that, a large
number of CHIKV cases have been notified in FS, which is the second-most populous city in
Bahia state, northeastern Brazil, and plays an important role on the spread to other Brazilian
states due to climate conditions and the abundance of competent vectors. To better under-
stand CHIKV dynamics in Bahia state, we generated 5 complete genome sequences from a
local outbreak raised in *Serraria Brasil*, a neighbourhood in FS, by next-generation sequenc-
ing using Illumina approach. Phylogenetic reconstructions revealed that the new FS
genomes belongs to the ECSA genotype and falls within a single strongly supported mono-
phyletic clade that includes other older CHIKV sequences from the same location, suggest-
ing the persistence of the virus during distinct epidemic seasons. We also performed minor
variants analysis and found a small number of SNPs per sample (b_29L and e_45SR = 16
SNPs, c_29SR = 29 and d_45PL and f_45FL = 21 SNPs). Out of the 93 SNPs found, 71 are

under accession numbers: MK159123, MK159124, MK159125, MK159126, MK159127.

**Funding:** This research was funded by CNPq grant number (440685/2016-8); CAPES grant number (88887.130716/2016-00), and by Horizon 2020 (PRES-005-FEX-17-4-2-33) to LCJA. The funders had no role in study design, data collection and analysis, decision to publish, or preparation of the manuscript.

**Competing interests:** The authors have declared that no competing interests exist.

synonymous, 21 are non-synonymous and one generated a stop codon. Although those mutations are not related to the increase of virus replication and/or infectivity, some SNPs were found in non-structural proteins which may have an effect on viral evasion from the mammal immunological system. These findings reinforce the needing of further studies on those variants and of continued genomic surveillance strategies to track viral adaptations and to monitor CHIKV epidemics for improved public health control.

## Introduction

Chikungunya virus (CHIKV) has emerged as a public health concern posing significant issues in tropical and subtropical regions [1]. Since 2004 it has been globally spread causing epidemics in more than 100 countries [2]. Four CHIKV lineages have been described: West African; East/Central/South African (ECSA); Asian; Indian Ocean Lineage (IOL) [3–5].

In Brazil, the first autochthonous cases of CHIKV were confirmed in September 2014 in the municipality of Oiapoque, Amapá state in the North of Brazil, followed by the city of Feira de Santana (FS), Bahia state in Northeast region, around seven days later [6]. By that time genomic analysis have identified the East/Central/South/Africa (ECSA) genotype for the first time in the Americas in FS and the municipality stood out in national scenario due to the large number of reported cases of the disease [6, 7].

Feira de Santana is an important city in Bahia state as it is surrounded by the biggest road network of the state, where thousands of passengers and freight vehicles transit, allowing a large flow of people favoring the introduction and spread of new viruses into the city and to other Brazilian regions [8].

Since 2014, Bahia state and specially FS have reported a large number of positive cases for CHIKV infection. Released data from the Brazilian surveillance health system (SINAN) indicates that Bahia state reported a total of 50,880 cases of chikungunya fever in 2016 and 1,524 chikungunya cases, until the 2019 25th epidemiological week (June/2019). In the same period FS reported more than 300 cases per 100 thousand habitants [9–10].

Here we report evidence of the persistence of CHIKV ECSA genotype and shed light on a localized outbreak raised in the *Serraria Brasil*, an upper medium class neighborhood within FS in 2016, two years after the lineage introduction in the locality.

## Materials and methods

### Ethics statement

This project was supported by the Pan American World Health Organization (PAHO) and the Brazilian Ministry of Health (MoH) as part of the arboviral genomic surveillance efforts of the ZiBRA project (www.zibraproject.org). Ethical approval for human samples was obtained from the Ethical Committee for Research from Gonçalo Moniz Institute, Oswaldo Cruz Foundation (IGM/FioCruz/BA) under CEP/CAAE number 45279715.8.0000.0040 and from Ethics Review Committee from PAHO under the reference number 2016-08-0029. Samples were provided for research and surveillance purposes within the terms of Resolution 510/2016 of CONEP (National Ethical Committee for Research, Ministry of Health).

### Study population

Blood, urine and saliva samples (n = 69) from 27 patients presenting symptoms consistent with CHIKV infection from Serraria Brasil neighborhood were collected by active surveillance

of Municipal Health Surveillance Division of Feira de Santana (SMS-SVS). Molecular diagnostics (RT-qPCR) were performed by Virology and Experimental Therapy Laboratory (LaVite) at Aggeu Magalhães Institute (IAM) FIOCRUZ-PE and specific CHIKV-IgG and CHIKV-IgM serology (ELISA- *Enzyme-Linked Immunosorbent Assay*) by Bahia state central public laboratory (LACEN-BA).

## Viral RNA isolation and sample processing

Viral RNA was extracted from 200µL of clinical samples using QIAmp Viral RNA Minikit (Qiagen) according to the manufacturer's instructions. Samples were linked to a digital record and clinical information such as date of onset of symptoms, sample collection date, municipality, state of residence, age, sex, residence type and when available, travel history.

## Real-time quantitative PCR

Reverse transcription quantitative real-time PCR (RT-qPCR) was performed on samples using the GoTaq® Probe 1-Step RT-qPCR System (PROMEGA) on an ABI7500 Real Time PCR Systems or a QuantStudio® Systems (Applied Biosystems). The CHIKV non-structural protein 1 (nsp1) was targeted using the primers CHIKV-F (5' to 3': AAAGGGCAAACTCAGCTT CAC), CHIKV-R (5' to 3': GCCCTGGGCTCATCGTTATTC) and the CHIKV Probe (5' to 3': FAM-CGCTGTGATACAGTGGTTTCGTGTG), based on an assay previously described [11]. Thermocycler conditions consisted of reverse transcription at 45°C for 15 mins followed by RT inactivation at 95° C for 2 mins, 40 cycles of denaturation at 95°C for 15 sec and annealing at 60° C for 1min.

## cDNA synthesis

All positive samples were submitted to a cDNA synthesis protocol using Protoscript II First Strand Sequencing kit (New England Biolabs—NEB). Then, a multiplex PCR was conducted using Q5 High Fidelity Hot-Start DNA Polymerase (New England Biolabs) and a sequencing primer scheme (divided into two separated pools) designed using Primal Scheme online tool to amplify 400 bp overlapping amplicons of the CHIKV complete genome (http://primal. zibraproject.org) [12]. All samples were subjected to 45 cycles of PCR using the thermocycling conditions of [12]. PCR products were purified using a 1x SPRI bead cleanup (Ampure XP Beads Agencourt) and concentrations were measured using a Qubit dsDNA High Sensitivity kit on a Qubit 3.0 fluorimeter (ThermoFisher).

## Library prep sequencing for Illumina

Nextera XT Sample Preparation Kit (Illumina Inc) was used to construct a DNA library for each sample using dual barcodes. After library preparation each samples was quantified using Nebnext® library quant (Illumina Inc) following the manufacturer's instructions and normalized in equimolar quantities before loading the flow cell. The library was deep-sequenced using the MiSeq Illumina platform with 2 x 75 bp paired ends, which allow us to sequence both ends of a fragment and generate high-quality alignable sequence. Paired-end reads were demultiplexed using the vendor software from Illumina. Demultiplexed Illumina reads were mapped on the KP164568 reference genome using Bowtie2 program with default parameters (https://www.ncbi.nlm.nih.gov/pmc/articles/PMC3322381/). Final consensus sequences were generated by the consensus module of Integrate Genome Viewer (https://www.ncbi.nlm.nih. gov/pmc/articles/PMC3346182/) with a 5x minimum read depth coverage. Any nucleotide variants on the primer regions were removed from the final consensus sequence.

## Phylogenetic analysis

Nucleotide sequences recovered from this study were first subtyped using Chikungunya TypingTool (https://genomedetective.com/app/typingtool/chikungunya/) [13]. New sequences were aligned to complete or almost complete reference CHIKV genome sequences (>10,000 bp), retrieved from National Center for Biotechnology Information (http://www.ncbi.nlm.nih.gov/) covering all four existing lineages. Reference strains were included based on the following criteria: 1) published in peer-reviewed journals; 2) no uncertainty regarding lineage assignment of each sequence; 3) non-recombinant classification using RDP4 recombination detection software. Alignment was performed using MAFFT online program [14] and manually edited by using AliView [15]. A maximum likelihood phylogeny was reconstructed from the concatenated dataset (n = 225) using IQ-TREE 1.6.8 software under the HKY nucleotide substitution model with 4 gamma categories (HKY+4G) which was inferred in jModelTest as the best fitting model [16]. Statistical robustness of tree topology was inspected using 100 bootstrap replicates [17]; bootstrap value >90% was considered statistically significant. From the ML generated using the concatenated dataset we selected all ECSA taxa from Brazil (ECSA-BR dataset) (n = 36) samples in different states Alagoas n = 25; Bahia n = 5; Paraiba n = 2; Pernambuco n = 1; Rio de Janeiro n = 2; Sergipe n = 1.

## Molecular clock phylogenetic analysis

In order to investigate the temporal signal in our CHIKV-ECSA dataset, we regressed root-to-tip genetic distances from this ML tree against sample collection dates using TempEst v 1.5.1 [18]. The ML phylogeny was used as a starting tree for Bayesian time-scaled phylogenetic analysis using BEAST 1.10.2 [19]. In the Bayesian analyses, we used an HKY+4G substitution model with a Bayesian skygrid coalescent model with 20 grid points [20]. We computed MCMC duplicate runs of 50 million states each, sampling every 5.000 steps for the ECSA-BR dataset. Convergence of MCMC chains was checked using Tracer v.1.7.1 [21]. Maximum clade trees were summarized from the MCMC samples using TreeAnnotator after discarding 10% as burn-in.

## Single Nucleotide Polymorphisms (SNPs) analysis

**Minority variants analysis.** SAMtools and bcftools packages (https://www.ncbi.nlm.nih.gov/pubmed/19505943, https://www.ncbi.nlm.nih.gov/pmc/articles/PMC3198575/) were used to perform variant calling on .bam files using the paramenters "mpileup -Bu -d 50000" and "call -O b -v -c -" respectively. Additionally, VCFtools (https://academic.oup.com/bioinformatics/article/27/15/2156/402296) was used to annotate the variants with the following parameters (—filter Qual = 20/MinDP = 200/SnpGap = 20). Finally, we used SnpEff [22] on the .vcf file to gather further insights on the effects of the SNPs found.

# Results

## Sample collection, qRT-PCR screening and sequencing

The study group was composed by 27 patients, 74% (n = 20) female and 26% (n = 7) male individuals, who exhibited CHIKV symptoms as intense polyarthralgia with impaired walking (n = 27), myalgia (n = 27), headache (n = 27), fever (n = 25), backache (n = 24), exanthema (n = 22), conjunctivitis (n = 17), retro-orbital pain (n = 16), nausea (n = 15) and vomiting (n = 14) (S1 Table).

The 69 clinical samples collected were typed as blood (serum or plasma, n = 27), urine (n = 21) and saliva (n = 21). CHIKV IgM serology was positive for 17 cases (73,91%; S1

**Table 1. Epidemiological data associated with isolates analysed in this study.**

| ID | Sample | Host | State | Municipality | Collection date | Sex | Age |
|---|---|---|---|---|---|---|---|
| FS144 | Blood (*plasma*) | Human | BA | Feira de Santana | 2016/10/03 | M | 32 |
| FS144 | Blood (*serum*) | Human | BA | Feira de Santana | 2016/10/03 | M | 32 |
| FS160 | Blood (*plasma*) | Human | BA | Feira de Santana | 2016/10/13 | F | 60 |
| FS160 | Blood (*serum*) | Human | BA | Feira de Santana | 2016/10/13 | F | 60 |
| FS160 | Saliva | Human | ES | Feira de Santana | 2016/10/13 | F | 60 |

ID = Project identifier; Sample = sample type; Host = Host species; Collection Date = Date of sample collection.

Table.), in two (11,76%) of those was possible to identify the presence of the virus genomic RNA through RT-qPCR on different cellular compartments (urine, blood and saliva). Samples tested by RT-qPCR showed cycle threshold values (Cts) ranging from 25,0 to 33,0 (Table 1). Median of RT-qPCR Cts for positive samples was 28.0, and was lower in blood (Ct 25, range: 25.0 to 28.0) than in urine (Ct 28.0) or saliva (Ct 33.0).

## Phylogenetic and molecular clock inference

To investigate and to better understand the diversity of CHIKV in some of most affected municipalities in Bahia state, we generated 5 CHIKV near-complete genomes (coverage range 88.90%-99.82%, mean = 97,4%) (Table 2) using next-generation sequencing technologies. A regression of genetic divergence from root to tip against sampling dates confirmed sufficient temporal signal ($r^2$ = 0.80). Phylogenetic analysis indicates that new generated sequences belong to the ECSA genotype (S1 Fig) which was detected for the first time in 2014 in Feira de Santana in the Bahia state. ML and Bayesian phylogenetic analyses revealed that the ECSA sequences from Serraria Brasil neighbourhood form a single well supported clade (bootstrap support = 1.0; posterior probability support = 1.0) (Fig 1). We estimated the date of the most recent common ancestor (tMRCA) of the Feira de Santana Clade to be around Mid-May 2016 (95% Bayesian credible intervals, BCI: June 2015 –February 2016, Fig 1), suggesting a local persistence of the virus in Feira de Santana across a period of 2 years during distinct epidemic seasons.

## Single Nucleotide Polymorphisms (SNPs)

We found a small number of SNPs per sample varying from 16 to 21 (b_29L and e_45SR = 16 SNPs, c_29SR = 29 and d_45PL and f_45FL = 21 SNPs). 71 out of 93 SNPs found are synonymous, 21 are non-synonymous and one generate a stop codon (Fig 2). Interestingly, 41 of all 93 SNPs detected are minor variants, that are supported by a substantial amount of reads but

**Table 2. Statistics for the 5 new CHIKV sequences generated in this study.**

| ID | Accession Number | Average Coverage Depth | Coverage Breadth (%) | Mapped million reads on KP164568 reference genome | Ct value |
|---|---|---|---|---|---|
| FS160 | MK159123 | 7646,74 | 99.26 | 1.794,160 | 28.0 |
| FS160 | MK159124 | 16673.25 | 99.37 | 3.035,895 | 25.0 |
| FS144 | MK159125 | 13284.30 | 99.69 | 2.314,812 | 28.0 |
| FS144 | MK159126 | 13316.89 | 99.82 | 2.368,027 | 28.0 |
| FS144 | MK159127 | 13049.04 | 88.90 | 2.090,492 | 33.0 |

Numbers correspond do coverage depth, coverage breadth and quantity of reads mapped on reference genome. ID = study identifier; Accession number = NCBI accession number; Ct = RT-qPCR quantification cycle threshold value

A. Spatial area under investigation:
Feira de Santana and its highways network.

B. Phylogenetic tree of CHIKV genomes from Feira de Santana

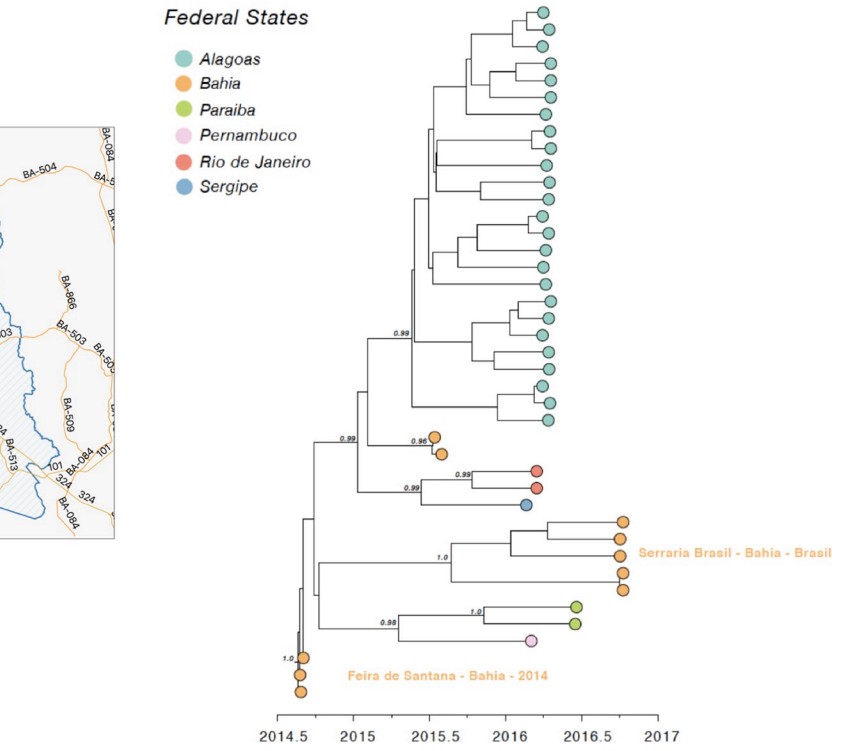

**Fig 1. Phylogenetic analysis of chikungunya virus human samples from Feira de Santana, Bahia, Brazil.** The municipality of Feira de Santana (FS) is located at a confluence of national highways. In A, federal highways BR-116 and BR-324 are shown in FS area. The BR-116 is the second longest highway in Brazil, it comprises 4,490 kilometers (2,790 mi) connecting Fortaleza (Ceará), one of the largest Northeast Brazil metropolises, to the southern city of Jaguarão, (Rio Grande do Sul), in the border with Uruguay. The BR-324 begins in Balsas (Maranhão) and ends in Salvador, where it plays an important role in connecting the road junction in FS to the capital, making it one of the main highways in the state. In B, new generated sequences belong to CHIKV-ECSA genotype and is clustered in a single strongly supported monophyletic clade that includes older FS sequences (bootstrap support = 98%) (orange).

in a lower proportion compared with the reference nucleotide (S2 Fig). Eight of these correspond to non-synonymous changes.

## Discussion

In this study, by performing Illumina approach sequencing, we generated 5 new CHIKV near-complete genomic sequences from 2016, collected in a neighbourhood in the municipality of Feira de Santana, Bahia state.

Our phylogenetic analysis showed that the novel genomes belong to ECSA genotype corroborating with previous studies [7, 23]. Although CHIKV is related to explosive outbreaks around the world [24–25], here we report a small local outbreak in *Serraria Brasil*, an upper medium class neighbourhood within FS.

Despite of the raising of outbreaks by new introductions of the virus into populations, our analysis shows that the novel sequences do not represent a re-introduction of the CHIKV into FS but confirm the basal circulation of the virus and its re-emergence in a local and susceptible population of FS, evidencing the persistence of the ECSA genotype in the region two years after its introduction.

The new genomes reported here were obtained from different cellular compartments of two CHIKV infected patients (Table 1). As reported in a previous study, blood, saliva and urine may also be used for the diagnosis of CHIKV infection, and the chances of detection are

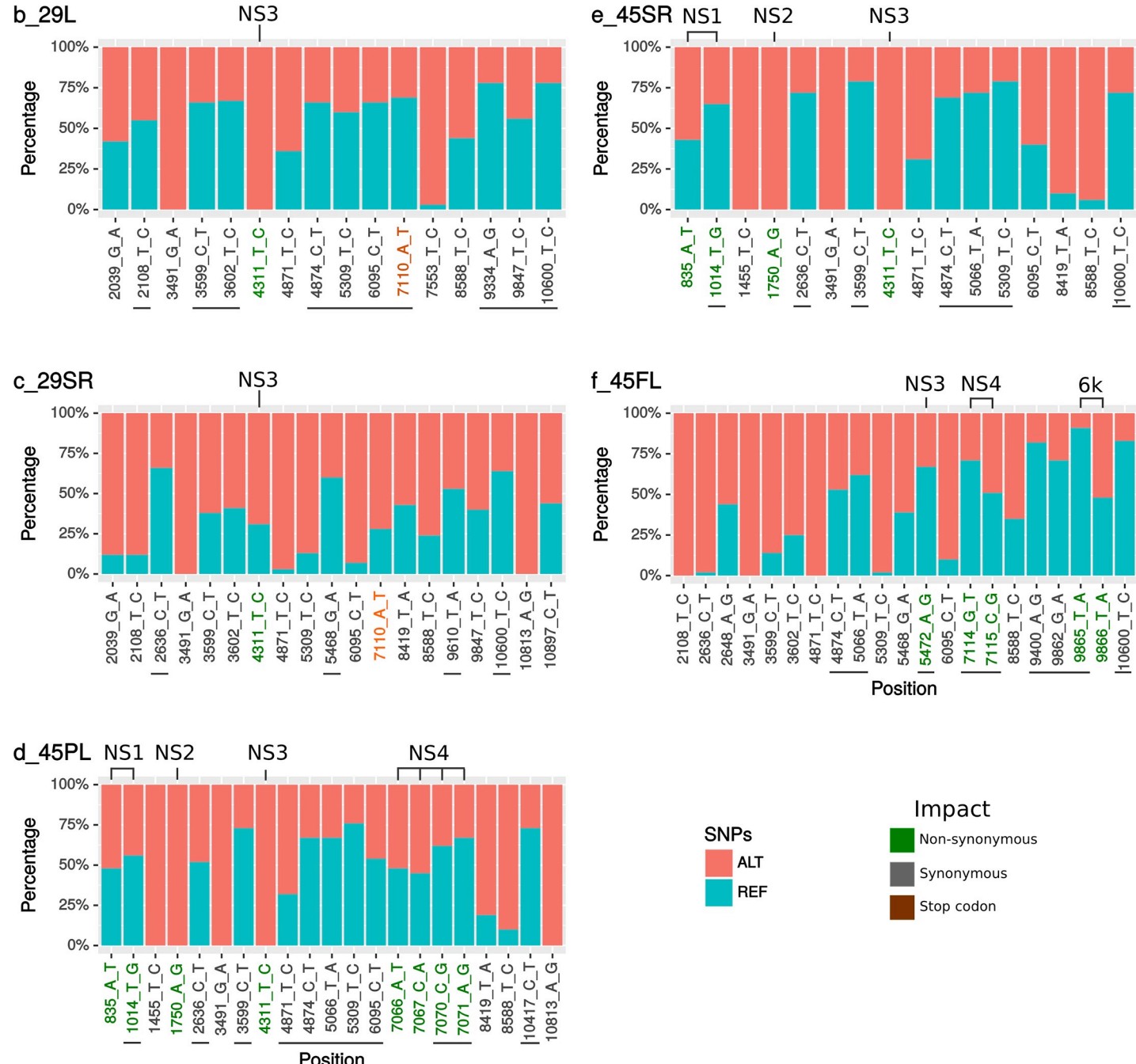

**Fig 2. Single Nucleotide Polymorphisms (SNPs) analysis.** Proportion of reads that supports each reference (blue) or SNP (red) variant. SNPs names denotes the position along the KP164568 reference genome following by the reference and variant nucleotide. Green, grey and brow SNPs names are non-synonymous, synonymous and stop codons SNPs. b_29L and c_29SR are blood from patient 1 (plasma and serum respectively); d_45PL correspond to plasma, e_45SR to serum and f_45FL to saliva from patient 2.

greatest when sampling occurs during the first week after the onset of symptoms [26]. These findings are particularly important for the genomic surveillance of arbovirus in regions with limited logistic structure, especially when the collection of blood samples, which is preferred, is not possible.

All 27 patients sampled in this study exhibited compatible symptoms for CHIKV infection. Taken together, these findings corroborate to other studies that demonstrate that 70% of CHIKV infection cases are symptomatic, since the virus rate of attack is high [27–28]. The difference of positive results between ELISA and RT-qPCR techniques is justified by the lack of time from the onset of symptoms and collection date performed, since the viremic peak–that would be detected by RT-qPCR–occurs in first days of infection [29, 23] unlike IgM antibodies levels that can be detected early as 5 days of infection up to 2 months [27, 29–31].

Localized outbreaks have been reported in other locations like the two small villages from Ravenna in Italy in 2007 [32] and more recently in Coutos neighbourhood of the city of Salvador, Bahia state, Brazil [33]. These outbreaks were related to high density of the mosquitoes, such as *Aedes albopictus* in the first study and *Culex quinquefasciatus* and *Aedes aegypti* in the second, although no CHIKV infected mosquito was reported. According to epidemiological surveillance data from *Aedes aegypti* Rapid Index Survey (LIRAa), the house index (HI) in FS was 2.27% in 2016 and 1.39% in 2017 [34]. The HI related to the infestation rates of *Aedes aegypti* mosquito [35] and provides qualified information for the municipalities to deploy arbovirus prevention and control strategies. According to Ministry of Health, HI values above 1% indicates risk of epidemics, thus, in 2016 FS was at risk of transmission of dengue and other arbovirus infections such as CHIKV [36] and may have had in impact on this outbreak.

The strategic location of the FS, at a road junction (Fig 1), where there is an intense movement of people from all Brazilian regions including other northeast cities, may further the circulation of infected patients or of subjects in the incubation period of arboviruses (DENV, CHIKV, ZIKV), that allied to climatic conditions and the density of *Aedes* mosquitoes [7], may contribute to virus dispersion within the region and beyond.

CHIKV infection in FS was characterized by two distinct epidemic waves (S3 Fig), the first one took place 3 months after the virus introduction by a returning traveller in 2014 and the second wave occurred in 2015 between 4th and 11th epidemiological weeks. Climate conditions and the HI related factors may have contributed for this epidemiological behaviour of CHIKV in FS. When the virus was introduced in July 2014, the climate conditions did not favor the reproduction and dispersion of *Aedes* mosquitoes (vector), although the population was immunologically susceptible [37]. On the other hand, a second epidemic wave occurred in a rain-intermittent period that contributes to urban and dwelling water accumulation and may have favored vector proliferation and expansion of infection. Also, the sub-notification of CHIKV cases by health care services may have masked the real range of the epidemy between the two waves [27,38].

The first epidemic wave initiated in George America neighborhood which reported the first cases of CHIKV in FS. That location represents the epicenter of the 2014 epidemy from where the infection expanded to surrounding neighborhoods [39]. Historically, the George America neighborhood is characterized by the low social status of its residents and by precarious sanitary conditions that might have favored the rapid dispersion of the CHIKV infection.

In contrast, Serraria Brasil neighborhood is placed far from the epicenter and is located in a region with better sanitary and environmental conditions since water supply and garbage collection services are provided more frequently by public services. We observed that along 2014 and 2015, the epidemic waves affected peripheral and more populous neighborhoods, while in 2016, when the epidemic has ceased, the reported cases predominate in less vulnerable neighborhoods, such as Serraria Brasil, where there was still susceptible population.

Regarding SNPs found in our analysis, previous studies have reported the occurrence of CHIKV mutations that modified its adaption to mosquito vectors such E1-A226V mutation, that increase IOL strain replication rate in *Aedes albopictus* [40–42]. We performed protein alignments to investigate the presence of the A226V (E1 protein) on the novel sequences

generate in our study but we did not observe it. Also, the non-synonymous SNPs found here are not related to any known CHIKV mutations that increase the virus replication and/or infectivity in vector or mammalian hosts. However, several non-synonymous mutations, both fixed and minor variant, were found in non-structural proteins which may have an effect on viral evasion from the mammal immunological system as reported by others [43–45]. These findings reinforce the need of further studies and continuous genomic surveillance to track viral adaptations and to identify main sources of transmission for improved public health actions, especially regarding vector control once the increase of mosquito-borne diseases is associated to the occurrence of their competent vectors in conjunction with adequate climate conditions [46].

In addition, genomic surveillance is a powerful tool to monitor virus adaptation to mosquitoes vectors, making possible the study of CHIKV fitness and evolution in mosquito populations, foretelling increase in viral infectivity and the risk of its emergence [47]. Also, by using complete or near complete viral genomes, spatial-temporal analysis can be performed to infer viruses introduction and dispersion events in the past. This approach was employed in previous studies and have shown evidences of cryptic transmission of arboviruses such as Zika, dengue and chikungunya before the first case detection [48–50].

On this way the combination of genomic surveillance with established surveillance strategies can be employed to help health laboratories in monitoring circulating viruses and to predict upcoming outbreaks heading public health actions such as the reorganization of the health care network, the implementation of health education actions, social mobilization and vector control [51].

Together, our results indicate the persistence of CHIKV ECSA lineage in the municipality of Feira de Santana and shed light to the risk of rise of a new localized outbreak. Our findings reinforce the needing of continuous genomic surveillance strategies and further studies on minor variants to track viral adaptations and to improve our understanding about CHIKV circulation in FS and to prevent new epidemics.

## Supporting information

**S1 Fig. Maximum-likelihood phylogenetic tree.**
(TIF)

**S2 Fig. Chikungunya virus genetic statistics.**
(TIFF)

**S3 Fig. Chikungunya notified cases by epidemiological week (SE) in Feira de Santana-BA, 2014–2016.**
(TIF)

**S1 Table. Clinical data of cases included in the study.**
(PDF)

## Acknowledgments

The authors thank all personnel from Health Surveillance System from Bahia that coordinated surveillance and helped with data collection and assembly. They also thank the ZiBRA2 project, the Brazilian Ministry of Health (SVS-MS), the Pan American Organization (OPAS) and the Virology and Experimental Therapy Laboratory (LaVite) at Aggeu Magalhães Institute (IAM) FIOCRUZ-PE that helped with laboratory and genome sequencing protocols.

## Author Contributions

**Conceptualization:** Jaqueline Goes de Jesus, Mariana Carolina Sobral, Rivaldo Venâncio da Cunha, Marta Giovanetti, Luiz Carlos Junior Alcantara.

**Data curation:** Jaqueline Goes de Jesus, Maricelia Lima Maia, Vagner Fonseca, Alvaro Salgado de Abreu, Helineide Ramos do Amaral, Siane Campos de Souza, Melissa Barreto Falcão, Erenilde Cerqueira, Tatiana Maria Teodoro Rezende, Mylena Ribeiro Pereira, Felicidade Mota Pereira, Zuinara Pereira Gusmão Maia, Rafael Freitas de Oliveira França, André Luiz de Abreu, Nuno Rodrigues Faria, Rivaldo Venâncio da Cunha, Marta Giovanetti, Luiz Carlos Junior Alcantara.

**Formal analysis:** Jaqueline Goes de Jesus, Joilson Xavier, Vagner Fonseca, Marta Giovanetti.

**Funding acquisition:** Luiz Carlos Junior Alcantara.

**Investigation:** Jaqueline Goes de Jesus, Gabriel da Luz Wallau, Maricelia Lima Maia, Joilson Xavier, Maria Aparecida Oliveira Lima, Vagner Fonseca, Alvaro Salgado de Abreu, Stephane Fraga de Oliveira Tosta, Helineide Ramos do Amaral, Italo Andrade Barbosa Lima, Paloma Viana Silva, Daiana Carlos dos Santos, Aline Sousa de Oliveira, Siane Campos de Souza, Melissa Barreto Falcão, Erenilde Cerqueira, Laís Ceschini Machado, Mariana Carolina Sobral, Tatiana Maria Teodoro Rezende, Mylena Ribeiro Pereira, Felicidade Mota Pereira, Zuinara Pereira Gusmão Maia, Rafael Freitas de Oliveira França, Carlos Frederico Campelo de Albuquerque e Melo, Rivaldo Venâncio da Cunha, Marta Giovanetti.

**Methodology:** Jaqueline Goes de Jesus, Gabriel da Luz Wallau, Vagner Fonseca, Italo Andrade Barbosa Lima, Paloma Viana Silva, Daiana Carlos dos Santos, Aline Sousa de Oliveira, Melissa Barreto Falcão, Erenilde Cerqueira, Laís Ceschini Machado, Mariana Carolina Sobral, Tatiana Maria Teodoro Rezende, Mylena Ribeiro Pereira, Zuinara Pereira Gusmão Maia, Rafael Freitas de Oliveira França, Rivaldo Venâncio da Cunha, Marta Giovanetti.

**Resources:** Maricelia Lima Maia, Maria Aparecida Oliveira Lima, Helineide Ramos do Amaral, Siane Campos de Souza, Melissa Barreto Falcão, Erenilde Cerqueira, Laís Ceschini Machado, Felicidade Mota Pereira, Zuinara Pereira Gusmão Maia, André Luiz de Abreu, Carlos Frederico Campelo de Albuquerque e Melo, Rivaldo Venâncio da Cunha, Luiz Carlos Junior Alcantara.

**Software:** Vagner Fonseca, Marta Giovanetti.

**Supervision:** Marta Giovanetti, Luiz Carlos Junior Alcantara.

**Validation:** Nuno Rodrigues Faria, Marta Giovanetti.

**Visualization:** Marta Giovanetti.

**Writing – original draft:** Jaqueline Goes de Jesus, Gabriel da Luz Wallau, Marta Giovanetti, Luiz Carlos Junior Alcantara.

**Writing – review & editing:** Jaqueline Goes de Jesus, Gabriel da Luz Wallau, Maricelia Lima Maia, Joilson Xavier, Maria Aparecida Oliveira Lima, Vagner Fonseca, Alvaro Salgado de Abreu, Stephane Fraga de Oliveira Tosta, Italo Andrade Barbosa Lima, André Luiz de Abreu, Carlos Frederico Campelo de Albuquerque e Melo, Nuno Rodrigues Faria, Rivaldo Venâncio da Cunha, Marta Giovanetti, Luiz Carlos Junior Alcantara.

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
