## [Decision Letter · Decision Letter 0]

1 Nov 2019

PONE-D-19-29145

Persistence of Chikungunya ECSA genotype and local outbreak in an upper medium class neighbourhood in Northeast, Brazil.

PLOS ONE

Dear Dr Luiz Carlos Junior Alcantara,

Thank you for submitting your manuscript to PLOS ONE. After careful consideration, we feel that it has merit but does not fully meet PLOS ONE’s publication criteria as it currently stands. Therefore, we invite you to submit a revised version of the manuscript that addresses the points raised during the review process.

We would appreciate receiving your revised manuscript by November 30th. To enhance the reproducibility of your results, we recommend that if applicable you deposit your laboratory protocols in protocols.io, where a protocol can be assigned its own identifier (DOI) such that it can be cited independently in the future. For instructions see: http://journals.plos.org/plosone/s/submission-guidelines#loc-laboratory-protocols

We look forward to receiving your revised manuscript.

Kind regards,

Massimo Ciccozzi

Academic Editor

PLOS ONE

Journal Requirements:

This work was supported by the ZiBRA2 project supported by the Brazilian Ministry of Health (SVS- MS) and the Pan American Organization (OPAS) and founded by Decit/SCTIE/MoH and CNPq (440685/2016-8 and 440856/2016-7); by CAPES (88887.130716/2016-00, 88881.130825/2016-00 and 88887.130823/2016-00); by EU’s Horizon 2020 Programme through ZIKAlliance (PRES-005-FEX-17-4-2-33). The Virology and Experimental Therapy Laboratory (LaVite) at Aggeu Magalhães Institute (IAM) FIOCRUZ-PE is supported by Fundação de Amparo à Ciência e Tecnologia do Estado de Pernambuco (FACEPE) [grant number APQ-0078-2.02/16].

LCJA: This research was funded by CNPq grant number (440685/2016-8); CAPES grant number (88887.130716/2016-00).

Reviewers' comments:

Reviewer's Responses to Questions

**Comments to the Author**

1. Is the manuscript technically sound, and do the data support the conclusions?

Reviewer #1: Yes

Reviewer #2: Yes

2. Has the statistical analysis been performed appropriately and rigorously? 

Reviewer #1: I Don't Know

Reviewer #2: Yes

3. Have the authors made all data underlying the findings in their manuscript fully available?

Reviewer #1: Yes

Reviewer #2: Yes

4. Is the manuscript presented in an intelligible fashion and written in standard English?

Reviewer #1: Yes

Reviewer #2: Yes

5. Review Comments to the Author

Reviewer #1: Goes de Jesus et al., in this research article report evidence of the persistence of CHIKV ECSA genotype and shed light on a local outbreak raised in the Serraria Brasil, an upper medium class neighbourhood within FS in 2016, two years after the lineage introduction in the locality.

Although there have now been many papers documenting various portions of the Chikungunya virus outbreak in Brazil, this manuscript benefits from the fact that no genomic data have been generated in 2016 in Bahia state, contributing to add knowledge regarding the persistence as well as the transmission dynamics of these virus in Brazil. This manuscript does not provide a large conceptual advance in our understanding of the Chikungunya virus outbreak in the Americas, but it does tell a nice story. With some minor changes, I think it would be suitable for publication in Plos One.

Comments:

- Please revise carefully the English used.

- What was the considerations for using 2 x 75 bp NGS reagent? Why not using larger read length such as 2 x 150 bp? Longer reads may decrease gaps.

- Figure 1. Is the zoom representing the city of Feira de Santana or the Serraria Brasil neighnorhood? Please state this and add the hype in the map.

- In the method section, could the authors add more information regarding the Library prep sequencing for Illumina, this is really fundamental in the field. Did the author used barcode? Have been this barcode pooled in an equimolar fashion after the amplification?

Reviewer #2: Goes de Jesus et al., in this research article, report evidence of the persistence of CHIKV ECSA genotype and shed light on a local outbreak raised in the Serraria Brasil, an upper medium class neighbourhood within FS in 2016, two years after the lineage introduction in the locality.

Bahia was the introductory point of CHIKV-ECSA in the Americas and it is the region in Brazil that seems to have the highest genetic diversity of this genotype. Since the first registered cases in 2014, no new genomic surveillance data had been released. Authors provide new genomic data from this state from 2016, and this is, in my point of view the main benefits of the present research article.

Despite these considerations I have a main concern:

The Authors state, in lines 347-349 that these data improve intervention strategies. How? Please provide strong evidence.

6. PLOS authors have the option to publish the peer review history of their article (what does this mean?). If published, this will include your full peer review and any attached files.

Reviewer #1: No

Reviewer #2: No

---

## [Author Response · Author response to Decision Letter 0]

14 Nov 2019

Response to reviewers

PONE-D-19-29145

Persistence of Chikungunya ECSA genotype and local outbreak in an upper medium class neighborhood in Northeast, Brazil.

In this document, we have addressed each review comment separately. We are glad that all editorial and review comments were acknowledged of significant value to our manuscript. We have considered all comments from the reviewers and the editorial board and we believe that the resulting manuscript conforms better with the journal’s guidelines, and its main messages are now clearer.

Journal requirements

Please remove any funding-related text from the manuscript and let us know how you would like to update your Funding Statement.

Answer: Thank you for this comment, we removed funding-related text from the manuscript (lines 380-385) and we will update our Funding Statement on the online submission form.

Reviewer #1

Please revise carefully the English used.

Answer: We agree with the reviewer and we have carefully revised the English used. We’ve made changes, specially converting British English to American English in order to meet journal requirements. This include a small change on “neighbourhood” word in the title which now is "Persistence of Chikungunya ECSA genotype and local outbreak in an upper medium class neighborhood in Northeast, Brazil” 

What was the considerations for using 2 x 75 bp NGS reagent? Why not using larger read length such as 2 x 150 bp? Longer reads may decrease gaps.

Answer: We agree with the reviewer about having a lower number of gaps when using longer reads, however it is most of the time true for de novo assembly. We used 2 x 75 just because that was the flowcell we had at the time of sequencing. Moreover, since we reference assembled a compact viral genome with no repetitive regions the small reads does not affected the assembly considerably. It can be observed on the high coverage breadth we obtained in our genomes. Lastly, the gaps were restricted to the 5’ and 3’ UTR regions that are more difficult to amplify and consequently to assembly, showing that the existing gaps was not due to the assembly problems because of the use of 75bp reads.

Figure 1. Is the zoom representing the city of Feira de Santana or the Serraria Brasil neighborhood? Please state this and add the hype in the map.

Answer: We thank the reviewer for this comment. 

In Figure 1 we show Feira de Santana municipality and the highways network that converge from all Brazilian regions in that place. We believe this is of particular importance for the dissemination of infectious diseases within the country, as the intense movement of people there, may contribute to virus dispersion to other locations. We added information about location on the map when it shows Brazil and the zoom that is showing Feira de Santana, highlighting the road junction.

In the method section, could the authors add more information regarding the Library prep sequencing for Illumina, this is really fundamental in the field. Did the author used barcode? Have been this barcode pooled in an equimolar fashion after the amplification?

Answer: We added more information in the material and methods section lines 136-149. Yes, we used barcodes specific to each sample which allowed us to separate the specific reads after sequencing.

Reviewer #2

The Authors state, in lines 347-349 that these data improve intervention strategies. How? Please provide strong evidence.

Answer: Thank you for your comment. We added discussion about how genomic surveillance can improve knowledge about mosquito-borne viruses and public health strategies (lines 372 – 390).

-

---

## [Editor Report · Decision Letter 1]

20 Nov 2019

Persistence of Chikungunya ECSA genotype and local outbreak in an upper medium class neighbourhood in Northeast, Brazil.

PONE-D-19-29145R1

Dear Dr. Luiz Carlos Junior Alcantara,

We are pleased to inform you that your manuscript has been judged scientifically suitable for publication and will be formally accepted for publication once it complies with all outstanding technical requirements.

With kind regards,

Massimo Ciccozzi

Academic Editor

PLOS ONE
---

## [Editor Report · Acceptance letter]

19 Dec 2019

PONE-D-19-29145R1 

Persistence of Chikungunya ECSA genotype and local outbreak in an upper medium class neighborhood in Northeast, Brazil 

Dear Dr. Alcantara:

I am pleased to inform you that your manuscript has been deemed suitable for publication in PLOS ONE. Congratulations! Your manuscript is now with our production department. 

With kind regards,

on behalf of

Prof Massimo Ciccozzi 

Academic Editor

PLOS ONE